# The Role of the Neutrophilic Network in the Pathogenesis of Psoriasis

**DOI:** 10.3390/ijms23031840

**Published:** 2022-02-06

**Authors:** Joanna Czerwińska, Agnieszka Owczarczyk-Saczonek

**Affiliations:** Dermatology, Sexually Transmitted Diseases and Clinical Immunology, School of Medicine, University Warmia and Mazury in Olsztyn, 10-719 Olsztyn, Poland; aganek@wp.pl

**Keywords:** psoriasis, neutrophil, extracellular trap, neutrophilic network

## Abstract

One role of neutrophils, the most abundant innate immune sentinels, is neutrophil extracellular trap (NET) formation, which plays a significant role in immune surveillance. However, NET operation is bidirectional. Recent studies report that NETs may contribute to the development of autoimmune diseases such as psoriasis. The participation of neutrophils in the pathogenesis of that disease is dependent on an autoinflammatory feedback loop between neutrophils, lymphocytes, dendritic cells and keratinocytes. Our aim was to clarify the field of NET research in psoriasis and highlight the main factors required for NET generation, which may be a target of new therapies. This article presents a comphrehensive review concerning studies addressing the participation of neutrophils in the pathogenesis of psoriasis. Based on the available English-language literature, we discuss original papers presenting significant research findings which may help to understand and interpret the NET formation process in psoriasis, as well as the newest systematic reviews on PubMed. Next, the comparison, synthesis and summary of reported results were performed to clearly indicate the specific component of the NET which participates in the development of psoriasis.

## 1. Neutrophils

At the beginning of infection, locally released molecules attract by chemotaxis the most abundant immune sentinels—phagocytes (e.g., neutrophils, macrophages). These phagocytes use a number of factors, including proteolytic enzymes and antibacterial peptides as well as reactive oxygen species (ROS) to engulf and dispose of microorganisms both inside cells and in the extracellular space [1].

In humans, neutrophils are classified into two heterogenous populations: low-density granulocytes (LDGs) located in the peripheral blood mononuclear cell (PBMC) fraction and typical polymorphonuclear neutrophils (PMNs) formed in a pellet together with erythrocytes [2].

With regard to neutrophils, the most characteristic specific granules include azurophilic, specific, gelatinase, secretory vesicles and the newly distinguished ficolin-1-rich granules [3,4]. Azurophilic granules mainly contain lysosomal proteins, i.a., myeloperoxidase (MPO) and the serine proteases neutrophil elastase (NE), proteinase 3 and cathepsin G, which are released into the extracellular space during degranulation. The migration and antimicrobial activity of neutrophils are caused by specific and gelatinase granules which secrete, i.a., lactoferrin, lipocalin, collagenase, cytochrome b558, MAC-1 and LL-37 cathelicidin [5,6]. The secretory vesicles promote neutrophil recruitment by releasing alkaline phosphatase [7,8] (Table 1). This process is related to the neutrophil extracellular trap’s (NET) formation by activated neutrophils [9].

NETs play a significant role in neutralizing intracellular and extracellular microorganisms, which conditions the release of specific factors: cytokines, chemokines and growth factors [1].

Extracellular traps were first described in neutrophils, but other types of granulocytes may also form traps. Eosinophil extracellular traps (EETs) are involved in protecting against parasites, and have a role in the allergic response [10]. Basophils (via BETs) may also externalize chromatin and kill bacteria in the absence of phagocytic activity [11,12]. Mast cells (via MCETs) also contribute to defense against some pathogens [13]. Extracellular traps may also be formed by monocytes or macrophages [14]. This process seems to function as an alternative defense mechanism that is activated when cell phagocytic capacity is insufficient [15].

## 2. Mechanism of Neutrophil Extracellular Traps (NETs)

The process of NET formation was first described in 2004, when it was classified as a unique form of cell death—NETosis [1]. It is currently known that NET creation is not always related to the death of cells, so using this name might be confusing.

Neutrophils may be activated in response to various microorganisms, such as bacteria, fungi and protozoa [16,17,18], but the induction of NETs does not need living pathogens and may be caused by various immune complexes, cytokines and chemokines, i.a., interferon (IFN), complement factor 5a [19], interleukin-8 (IL-8), tumor necrosis factor α (TNFα), phorbol-12-myristate-13-acetate (PMA), lipopolysaccharide (LPS), monosodium urate crystals and nitric oxide [1,20,21]. After reaching the site of infection, neutrophils may extrude large amounts of chromatin fibers decorated with over 30 types of granular cytoplasmic proteins, which trap and kill the microorganisms [1,3,4,5,6,7,8].

The process of NET formation is divided into two types: rapid (5–60 min) non-lytic NETosis (vital NETosis), which occurs after exposure to an infectious subject after phagocytosis and chemotaxis, where neutrophils are able to continue their functions; and slow (2–4 h) lytic NETosis (suicidal NETosis), occurring after hours of stimulation and connected with programmed cell death [22]. Suicidal NET formation is activated by different pathways, i.a., involving MPO and NE secretion [3,4,5,6,7,8].

There are several stages in NET creation (Figure 1). The first is chromatin decondensation, where positively charged arginines located in the histone are replaced by citrulline (with charge changing). This process, named histone deimination (or citrullination), is activated by protein arginine deiminase 4 (PAD-4, a Ca^2+^-dependent enzyme) [23]. This phenomenon is accompanied by ROS production caused by the activation of nicotinamide adenine dinucleotide phosphate (NADPH) oxidase, which requires the phosphorylation of the cytosolic subunits (p47-phox, p40-phox, p67-phox and Rac) to enable the combination with cytochrome b558 [24]. ROS are also particularly important in the antimicrobial activity of neutrophils. As a result, the membranes of the nucleus and granularity are lost, and the changed chromatin fiber may combine with proteins to form the structure released into the extracellular space [24]. It was shown that three granular proteins were the key molecules in the decondensation process: NE causing histone degradation, MPO changing chromatin structure (the distinction between euchromatin and heterochromatin is lost) and cathelicidin antimicrobial peptide (LL-37) connecting histones with DNA [25].

NET operation is bidirectional. On one hand, the protective activity of NETs based on the direct contact of trapped microorganisms with bactericidal components such as histones, granules released via degranulation and negatively charged DNA has been widely reported. Moreover, NETs promote relief from inflammation and wound healing by accumulating inflammatory mediators, and by degrading pro-inflammatory cytokines and chemokines [26]. Conversely, numerous reports have discussed the formation of dangerous aggregates of high neutrophil densities with wastes such as dead epithelial cells and bacteria, which might lead to vessel and duct clogging [27,28]. Additionally, the overexpression or insufficient removal of ROS may cause various dysfunctions associated with oxidative stress. It should also be noted that large amounts of NETs released in pathological conditions that cannot be effectively removed by DNases may aggravate inflammation, disrupt microcirculation and cause tissue damage. The release of large amounts of DNA, hormones and proteins stored in cells may also contribute to the development of autoimmune disorders [29]. Moreover, it was described that DNase might damage host tissues by the aggressive removal of non-phagocytic bacterial particle attachment [30].

In summary, neutrophils play a superior role in immune surveillance, but abundant reports suggest that NETs play an important role in the pathogenesis of autoimmune diseases such as psoriasis.

## 3. Psoriasis

The main causes of psoriatic skin lesions are keratinocyte hyper-proliferation and increased endothelial proliferation [31,32,33]. According to the latest research, psoriasis is classified as a chronic disease with systemic inflammatory features associated both with the skin and internal organs. Additionally, psoriasis is presented as an immunogenetic disorder resulting from the interactions between the innate and adaptive immune systems [34,35,36].

Psoriasis is triggered by a variety of inherited and acquired factors as well as external immunological substances such as interleukins [37,38]. The pathogenesis of psoriasis follows three different mechanisms: Th1/Th17/Th22 type-dominant cytokine imbalance, plasmacytoid dendritic cell (pDC) activation via toll-like receptors (TLRs) and interferon-α (INF-α) release. The latest study showed that some mechanical signals, followed by a complex of LL-37 and DNA/RNA, might also induce the disease. The activated complex of LL-37-DNA/RNA is incorporated into pDCs or conventional dendritic cells (cDCs—previously called myeloid DCs (mDCs)), where it is responsible for the Th1/Th17/Th22 differentiation of naive T cells [38,39,40].

What role do neutrophils play in the presented mechanism? Neutrophils are critical to the development of disease [41]. The course of psoriasis is usually characterized by a high accumulation of neutrophils both in plaques as well as in blood [42], and a decrease of circulating neutrophils is accompanied by the regression of psoriatic plaque [41]. Additionally, numerous biological therapies indirectly affect the function and number of those cells [43,44]. Regrettably, the role of neutrophils and NETs in the pathogenesis of psoriasis still requires in-depth scrutiny.

## 4. Neutrophils and NETs in Psoriasis

NETs are involved in each phase of psoriasis development, where they are able to initiate immune response through the creation of T-cell imbalance, keratinocyte proliferation, angiogenesis and the formation of autoantigens Discussing autoimmune diseases, the discovery of two neutrophil populations—of which LDGs are more effective at generating NETs than PMNs—seems to be particularly important. Patients with psoriasis showed increased levels of circulating LDG subsets. The quantity of circulating LDGs corresponded with the intensity of psoriatic lesions in [45]. LDGs also display an increased ability to synthesize TNFα or IFNs [46]. The latest research by Skrzeczynska-Moncznik et al. [47] confirmed the heterogeneity of neutrophils by revealing differences in the structure of the serine protease NE.

### 4.1. LL-37

The most significant role in the development of psoriasis is assigned to the LL-37 peptide that binds nucleic acids (DNA or RNA) and is overexpressed both in the psoriatic blood and skin. Moreover, the presence of circulating LL-37-specific T cells is correlated with psoriasis severity [37,42,48,49]. LL-37/DNA complex (resistant to degradation by nuclease) is detected by pDCs [37,50,51] and then transported to the endosome where it activates the toll-like receptor TLR9 [51,52] and cytosolic GAS/STING receptors [30,53]. This results in the secretion of IFN and activation of the inflammatory pathway and, therefore, acts on Th17 cells [37,40,51,54]. In turn, the cDCs are triggered in two direct ways: by IFN produced by pDCs, and by the LL-37/RNA complex. Furthermore, LL-37 induces the excessive production of pro-inflammatory cytokines in the TLR7-containing endosome, including IL-23, IL-12 and TNFα. IL-23 induces Th1 to produce IFN, whereas IL-12 and TNFα are responsible for the differentiation of Th17/Th22 cells and stimulation of IL-17 and IL-22 secretion [40,52]. Th1/Th17/Th22 cells specific to LL-37 migrate to the epidermis, where they cause the hyperproliferation of keratinocytes [40,55,56], which promote the development of psoriatic lesions and inflammation (Figure 2) [57]. This mechanism and the key role of TNF/IL-23/IL-17 signal axis in psoriasis was confirmed by the effectiveness of therapies which were anti-IL-12, anti-IL-17, anti-IL-23 and anti-TNFα [40,58] or their receptors [59].

Additionally, LL-37 enhances the secretion of IL-8 after the activation of p38 mitogen-activated protein kinases (MAPK p38) and extracellular signal-regulated kinase (ERK) [60]. LL-37 induces the release of human β-defensin 2 (HBD-2). Thus, the overexpression of LL-37 is related to low susceptibility to skin infections [33,60]. Furthermore, it was demonstrated that LL-37 had anti-apoptotic effects on keratinocytes, causing increased cellular proliferation [29,61]. In addition, it was reported that LL-37 might activate monocytes (P2 × 7 receptor), leading to inflammasome activation and IL-1β production [5,62], whereas DNA bound by LL-37 did not enhance Il-1β secretion in psoriasis [63]. For many years, only LL-37/DNA complex activating pDCs was considered to be the main mechanism of psoriasis. However, a study conducted by Herster et al. in 2020 revealed that RNA, both from neutrophils as well as foreign organisms (e.g., bacterial or fungal), might develop the immunostimulatory properties of LL-37 in psoriatic skin. It was demonstrated that neutrophils mounted self-propagating NETs and a cytokine response contributing to chronic inflammation in psoriasis [64].

### 4.2. Serine Proteases

NE, together with other serine proteases (i.a., MPO, cathepsin G and proteinase 3) released in the NET formation process, stimulates the production of ROS. Subsequently, pDCs and cDCs present antigens to T cells, resulting in the upset of Th1 and Th2 helper cell balance, intensive keratinocyte proliferation (by activating the growth factor receptor pathway—EGFR) and promotion of angiogenesis. ROS are additionally an activator of mitogen-activated protein kinase (MAPK), NF-κB or Janus kinase converter and activator of the developmental transcription protein (JAK-STAT) [65,66]. Moreover, proteases may trigger cDC response by the activation of the intracellular toll-like TRL4 receptor and cleave IL-36 precursors, leading to the exacerbation of the local inflammation of psoriatic tissue, for example by the continuous recruitment of neutrophils via interaction with keratinocytes producing CXCL1/2 and CXCL8 [67] and activating resting TNFα in the epithelial cell membrane (mTNFα) to the state of soluble TNFα (sTNFα) [68]. Elevated levels of NE and MPO were noted both in the serum and lesional skin of psoriatic patients [33,69]. The role of the gene encoding MPO is especially discussed in the development of generalized pustular psoriasis (GPP). Mutations in the MPO gene affect changes in neutrophil activity [70]. As a result of mutation, MPO deficiency causes the defective formation of NETs and the reduced phagocytosis of neutrophils by monocytes. Additionally, it was noted that the activity of MPO was inversely correlated with NE and proteinase 3, which regulate the IL-36 pathway [71].

### 4.3. Peptidyl Arginine Deiminase 4 (PAD-4)

Although numerous studies have been performed in different diseases, both in human and animal models, data concerning the involvement of PAD-4 in NET formation remains contradictory. On the one hand, it was demonstrated that PAD-4-mediated NETs were not required for disease development. On the contrary, some reports showed that the citrullination of NET components was essential [72,73,74,75,76]. Moreover, there is still a lack of research on the role of PAD-4 in the formation of NETs in human psoriasis. Interestingly, our preliminary study concerning the contribution of PAD-4 in patients with psoriasis showed an elevated serum protein level, which was correlated with the severity of the disease. It seems that PAD-4 may be considered a psoriasis marker (with TNFα and IL-8). Moreover, PAD-4 levels responded to the applied local and systemic therapies (data unpublished). Further research is needed to determine the role that PAD-4 plays in the pathogenesis of psoriasis.

### 4.4. Additional Triggers of NETs

The process of creating NETs is additionally regulated by external factors, including secretory leukocyte protease inhibitor (SLPI). It is a cationic protein that binds to DNA and NE and participates in the regulation of pDCs, which was confirmed by studies on humans and mice [38,77,78]. In addition, the formation of NETs is also controlled by eosinophils, regulating the pDC response through stored RNA. Eosinophils block the secretion of IFN-α and, thus, inhibit the activity of T cells and are involved in the regulation of IL-17 and IL-23 secretion [39]. Additionally, lipocalin-2 (LCN2), secreted by granulocytes and epidermal keratinocytes, seems to play a special role, as it regulates chemotaxis and NET formation. Increased levels of LCN2 were observed in the blood of patients with pustular psoriasis [79]. In addition, mRNA and protein studies demonstrated a correlation between LCN2 and IL-1β, IL-17 and TNFα levels [80]. With regards to the pathogenesis of psoriasis, an autoinflammatory feedback loop has been observed between neutrophils, lymphocytes and keratinocytes. NETs take part in the activation of Th17, while T cells expressing IL-17 produce cytokines regulating neutrophil recruitment, activation and survival [81]. In turn, the stimulation of keratinocytes leads to the secretion of chemokines (e.g., IL-8) and the antimicrobial peptide LL-37, inducing NET formation. Genetic research showed the key role of the TRAF3IP2 gene as the basis for the increased expression of IL-17 in psoriasis [82]. TRAF3IP2 encodes the D10N Act1 adapter molecule, which is a mediator of IL-17 signal transduction by the regulation of the transcription activator 3 (STAT3) of Th17 cells. The latest reports confirmed the stimulatory effects of NETs on the increase of T helper 17 cells in psoriasis. This process was also controlled by the variant of the TRAF3IP2 gene, which is associated with increased risk of psoriasis [81].

## 5. Summary

Neutrophils are the most important cells in the innate immune system, protecting the body from infections by microorganisms. However, recent studies have paid more attention to extracellular traps, which may damage the tissues of the host and may be involved in the progression of autoimmune disorders such as psoriasis. The participation of neutrophils in the course of psoriasis is dependent on an interaction with keratinocytes, dendritic cells and T cells. The increased amount of neutrophils and NET ingredients are present both in lesioned skin and in the serum of psoriatic patients. Thus, the influence of NETs is observed during every step of the development of psoriasis. The enhanced expression of NETs causes the intensive secretion of both IL-17 and inflammatory mediators, which consequently results in a multiplied amount of neutrophils. The results of recent studies identified a combined influence of NETs and genetic conditions in Th17 induction. Regrettably, knowledge of the neutrophil mechanism in psoriasis is still incomplete and further studies are needed in order to develop new therapies directed at the NET formation process without damaging neutrophils.

## Figures and Tables

**Figure 1 ijms-23-01840-f001:**
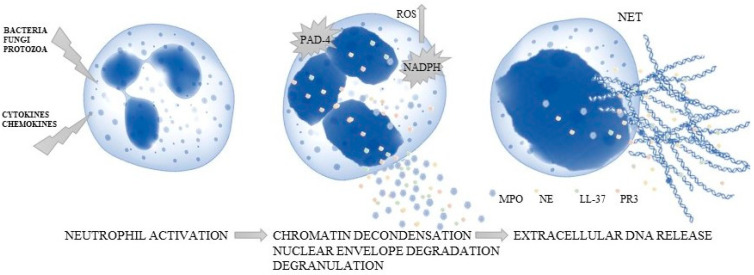
The process of NET creation.

**Figure 2 ijms-23-01840-f002:**
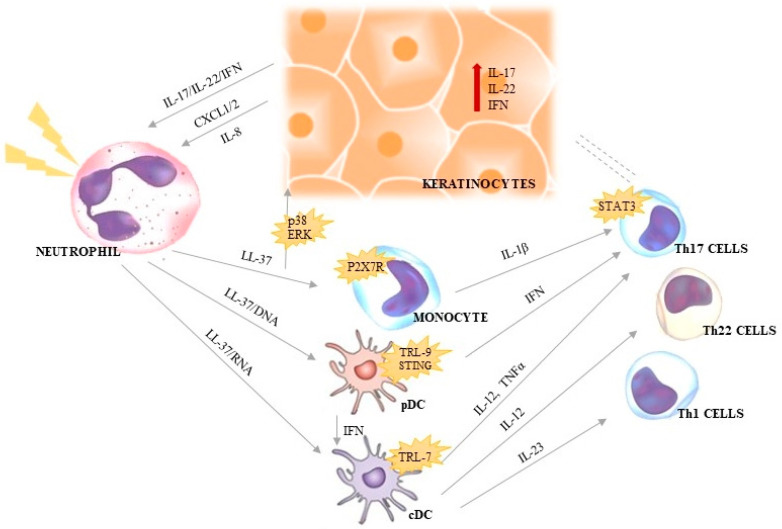
The autoinflammatory feedback loop between neutrophils, dendritic cells, lymphocytes and keratinocytes in psoriasis.

**Table 1 ijms-23-01840-t001:** Proteins released from activated neutrophils during NET creation.

Type of Granules	Other Name	Type of Proteins	Function of Proteins
Azurophilic	Peroxidase-positive or primary	myeloperoxidase, defensins, lysozyme, alkaline phosphatase, hydrolases, phospholipases (A2,C,D), bacterial permeability increasing protein, peroxidase 3, elastase, cathepsin G	antimicrobial activity
Specific	secondary	lactoferrin, lysozyme, alkaline phosphatase, NADPH oxidase, cathelicidin, collagenase	migration, antimicrobial activity
Gelatinase	tertiary	cathepsin, gelatinase, lipocalin, collagenase, cytochrome b558, ficolin	exocytosis, extracellular matrix degradation
Secretory	-	alkaline phosphatase	neutrophil recruitment (early stages of inflammatory response)
Ficolin-1 rich (gelatinase poor)	-	ficolin	lectin-initiated complement pathway

## Data Availability

Not applicable.

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
