# Peer review of "The Role of the Neutrophilic Network in the Pathogenesis of Psoriasis"

_ijms, 2022, doi:10.3390/ijms23031840_

Round 1

Reviewer 1 Report

Authors reported a narrative review with the aim of elucidating the role of NETs in psoriasis. The article is well written and referenced, the structure is logic and the figures seem to show a high quality. The findings that authors provided sound well, but I would like to put forward several questions to discuss.

  1. Methodology. Please, create e separate paragraph about methodological approaches which have been applied for searching paper, determining their quality, and reporting interpretation.
  2. Authors reported thoroughly and comprehensive description of the pathogenesis of the disease along with a key molecula - IL17 - which is a target for biological drugs implemented in the therapy of the disease. Please, extend the description of the role of several triggers in the pathogenesis of psoriasis so that the possibilities of biological and translational therapies in this aspect would sound better.

Author Response

We are grateful to the Reviewer for all valuable comments which helped us improve the manuscript. The all comments have been addressed and the manuscript has been revised accordingly.

According to the Reviewer’s suggestion, additional part about methods was prepared (line 18-25). The paragraph (line 152-168) was re-written to add information about other (than IL-17) triggers participating in development of psoriasis. The figure 2 was also changed. 

Reviewer 2 Report

This manuscript is very well prepared, and organized. Figures are adegate to the text. I think part of text may be good to convert to one more Table or pixture.

  Thank you

Author Response

We are grateful to the Reviewer for all valuable comments which helped us improve the manuscript. The all comments have been addressed and the manuscript has been revised accordingly.

According to the Reviewer’s suggestion, additional table (Tab.1., line 48) was prepared.
